# Generalization in Adaptive Data Analysis and Holdout Reuse*

**Cynthia Dwork**
Microsoft Research

**Vitaly Feldman**
IBM Almaden Research Center†

**Moritz Hardt**
Google Research

**Toniann Pitassi**
University of Toronto

**Omer Reingold**
Samsung Research America

**Aaron Roth**
University of Pennsylvania

## Abstract

Overfitting is the bane of data analysts, even when data are plentiful. Formal approaches to understanding this problem focus on statistical inference and generalization of individual analysis procedures. Yet the practice of data analysis is an inherently interactive and adaptive process: new analyses and hypotheses are proposed after seeing the results of previous ones, parameters are tuned on the basis of obtained results, and datasets are shared and reused. An investigation of this gap has recently been initiated by the authors in [7], where we focused on the problem of estimating expectations of adaptively chosen functions.

In this paper, we give a simple and practical method for reusing a holdout (or testing) set to validate the accuracy of hypotheses produced by a learning algorithm operating on a training set. Reusing a holdout set adaptively multiple times can easily lead to overfitting to the holdout set itself. We give an algorithm that enables the validation of a large number of adaptively chosen hypotheses, while provably avoiding overfitting. We illustrate the advantages of our algorithm over the standard use of the holdout set via a simple synthetic experiment.

We also formalize and address the general problem of data reuse in adaptive data analysis. We show how the differential-privacy based approach given in [7] is applicable much more broadly to adaptive data analysis. We then show that a simple approach based on description length can also be used to give guarantees of statistical validity in adaptive settings. Finally, we demonstrate that these incomparable approaches can be unified via the notion of approximate max-information that we introduce. This, in particular, allows the preservation of statistical validity guarantees even when an analyst adaptively composes algorithms which have guarantees based on either of the two approaches.

## 1 Introduction

The goal of machine learning is to produce hypotheses or models that generalize well to the unseen instances of the problem. More generally, statistical data analysis is concerned with estimating properties of the underlying data distribution, rather than properties that are specific to the finite data set at hand. Indeed, a large body of theoretical and empirical research was developed for ensuring generalization in a variety of settings. In this work, it is commonly assumed that each analysis procedure (such as a learning algorithm) operates on a freshly sampled dataset – or if not, is validated on a freshly sampled *holdout* (or testing) set.

Unfortunately, learning and inference can be more difficult in practice, where data samples are often reused. For example, a common practice is to perform feature selection on a dataset, and then use the features for some supervised learning task. When these two steps are performed on the same dataset, it is no longer clear that the results obtained from the combined algorithm will generalize. Although not usually understood in these terms, "Freedman's paradox" is an elegant demonstration of the powerful (negative) effect of adaptive analysis on the same data [10]. In Freedman's simulation, variables with significant $t$-statistic are selected and linear regression is performed on this adaptively chosen subset of variables, with famously misleading results: when the relationship between the dependent and explanatory variables is non-existent, the procedure overfits, erroneously declaring significant relationships.

Most of machine learning practice does not rely on formal guarantees of generalization for learning algorithms. Instead a dataset is split randomly into two (or sometimes more) parts: the training set and the testing, or holdout, set. The training set is used for learning a predictor, and then the holdout set is used to estimate the accuracy of the predictor on the true distribution (Additional averaging over different partitions is used in cross-validation.). Because the predictor is independent of the holdout dataset, such an estimate is a valid estimate of the true prediction accuracy (formally, this allows one to construct a confidence interval for the prediction accuracy on the data distribution). However, in practice the holdout dataset is rarely used only once, and as a result the predictor may not be independent of the holdout set, resulting in overfitting to the holdout set [17, 16, 4]. One well-known reason for such dependence is that the holdout data is used to test a large number of predictors and only the best one is reported. If the set of all tested hypotheses is known and independent of the holdout set, then it is easy to account for such multiple testing.

However such static approaches do not apply if the estimates or hypotheses tested on the holdout are chosen adaptively: that is, if the choice of hypotheses depends on previous analyses performed on the dataset. One prominent example in which a holdout set is often adaptively reused is hyperparameter tuning (*e.g.*[5]). Similarly, the holdout set in a machine learning competition, such as the famous ImageNet competition, is typically reused many times adaptively. Other examples include using the holdout set for feature selection, generation of base learners (in aggregation techniques such as boosting and bagging), checking a stopping condition, and analyst-in-the-loop decisions. See [13] for a discussion of several subtle causes of overfitting.

The concrete practical problem we address is how to ensure that the holdout set can be reused to perform validation in the adaptive setting. Towards addressing this problem we also ask the more general question of how one can ensure that the final output of adaptive data analysis generalizes to the underlying data distribution. This line of research was recently initiated by the authors in [7], where we focused on the case of estimating expectations of functions from i.i.d. samples (these are also referred to as statistical queries). .

## 1.1 Our Results

We propose a simple and general formulation of the problem of preserving statistical validity in adaptive data analysis. We show that the connection between differentially private algorithms and generalization from [7] can be extended to this more general setting, and show that similar (but sometimes incomparable) guarantees can be obtained from algorithms whose outputs can be described by short strings. We then define a new notion, *approximate max-information*, that unifies these two basic techniques and gives a new perspective on the problem. In particular, we give an adaptive composition theorem for max-information, which gives a simple way to obtain generalization guarantees for analyses in which some of the procedures are differentially private and some have short description length outputs. We apply our techniques to the problem of reusing the holdout set for validation in the adaptive setting.

**A reusable holdout:** We describe a simple and general method, together with two specific instantiations, for *reusing* a holdout set for validating results while provably avoiding overfitting to the holdout set. The analyst can perform any analysis on the training dataset, but can only access the holdout set via an algorithm that allows the analyst to validate her hypotheses against the holdout set. Crucially, our algorithm prevents overfitting to the holdout set even when the analyst's hypotheses are chosen adaptively on the basis of the previous responses of our algorithm.

Our first algorithm, referred to as Thresholdout, derives its guarantees from differential privacy and the results in [7, 14]. For any function $\phi : \mathcal{X} \to [0,1]$ given by the analyst, Thresholdout uses the holdout set to validate that $\phi$ does not overfit to the training set, that is, it checks that the mean value of $\phi$ evaluated on the training set is close to the mean value of $\phi$ evaluated on the distribution $\mathcal{P}$ from which the data was sampled. The standard approach to such validation would be to compute the mean value of $\phi$ on the holdout set. The use of the holdout set in Thresholdout differs from the standard use in that it exposes very little information about the mean of $\phi$ on the holdout set: if $\phi$ does not overfit to the training set, then the analyst receives only the confirmation of closeness, that is, just a single bit. On the other hand, if $\phi$ overfits then Thresholdout returns the mean value of $\phi$ on the training set perturbed by carefully calibrated noise.

Using results from [7, 14] we show that for datasets consisting of i.i.d. samples these modifications provably prevent the analyst from constructing functions that overfit to the holdout set. This ensures correctness of Thresholdout's responses. Naturally, the specific guarantees depend on the number of samples $n$ in the holdout set. The number of queries that Thresholdout can answer is exponential in $n$ as long as the number of times that the analyst overfits is at most quadratic in $n$.

Our second algorithm SparseValidate is based on the idea that if most of the time the analyst's procedures generate results that do not overfit, then validating them against the holdout set does not reveal much information about the holdout set. Specifically, the generalization guarantees of this method follow from the observation that the transcript of the interaction between a data analyst and the holdout set can be described concisely. More formally, this method allows the analyst to pick any Boolean function of a dataset $\psi$ (described by an algorithm) and receive back its value on the holdout set. A simple example of such a function would be whether the accuracy of a predictor on the holdout set is at least a certain value $\alpha$. (Unlike in the case of Thresholdout, here there is no need to assume that the function that measures the accuracy has a bounded range or even Lipschitz, making it qualitatively different from the kinds of results achievable subject to differential privacy). A more involved example of validation would be to run an algorithm on the holdout dataset to select an hypothesis and check if the hypothesis is similar to that obtained on the training set (for any desired notion of similarity). Such validation can be applied to other results of analysis; for example one could check if the variables selected on the holdout set have large overlap with those selected on the training set. An instantiation of the SparseValidate algorithm has already been applied to the problem of answering statistical (and more general) queries in the adaptive setting [1].

We describe a simple experiment on synthetic data that illustrates the danger of reusing a standard holdout set, and how this issue can be resolved by our reusable holdout. The design of this experiment is inspired by Freedman's classical experiment, which demonstrated the dangers of performing variable selection and regression on the same data [10].

**Generalization in adaptive data analysis:** We view adaptive analysis on the same dataset as an execution of a sequence of steps $\mathcal{A}_1 \to \mathcal{A}_2 \to \cdots \to \mathcal{A}_m$. Each step is described by an algorithm $\mathcal{A}_i$ that takes as input a fixed dataset $S = (x_1, \ldots, x_n)$ drawn from some distribution $\mathcal{D}$ over $\mathcal{X}^n$, which remains unchanged over the course of the analysis. Each algorithm $A_i$ also takes as input the outputs of the previously run algorithms $\mathcal{A}_1$ through $\mathcal{A}_{i-1}$ and produces a value in some range $\mathcal{Y}_i$. The dependence on previous outputs represents all the adaptive choices that are made at step $i$ of data analysis. For example, depending on the previous outputs, $\mathcal{A}_i$ can run different types of analysis on $S$. We note that at this level of generality, the algorithms can represent the choices of the data analyst, and need not be explicitly specified. We assume that the analyst uses algorithms which *individually* are known to generalize when executed on a fresh dataset sampled independently from a distribution $\mathcal{D}$. We formalize this by assuming that for every fixed value $y_1, \ldots, y_{i-1} \in \mathcal{Y}_1 \times \cdots \times \mathcal{Y}_{i-1}$, with probability at least $1 - \beta_i$ over the choice of $S$ according to distribution $\mathcal{D}$, the output of $\mathcal{A}_i$ on inputs $y_1, \ldots, y_{i-1}$ and $S$ has a desired property relative to the data distribution $\mathcal{D}$ (for example has low generalization error). Note that in this assumption $y_1, \ldots, y_{i-1}$ are fixed and independent of the choice of $S$, whereas the analyst will execute $\mathcal{A}_i$ on values $\boldsymbol{Y}_1, \ldots, \boldsymbol{Y}_{i-1}$, where $\boldsymbol{Y}_j = \mathcal{A}_j(S, \boldsymbol{Y}_1, \ldots, \boldsymbol{Y}_{j-1})$. In other words, in the adaptive setup, the algorithm $\mathcal{A}_i$ can depend on the previous outputs, which depend on $S$, and thus the set $S$ given to $\mathcal{A}_i$ is no longer an independently sampled dataset. Such dependence invalidates the generalization guarantees of individual procedures, potentially leading to overfitting.

*Differential privacy:* First, we spell out how the differential privacy based approach from [7] can be applied to this more general setting. Specifically, a simple corollary of results in [7] is that for

a dataset consisting of i.i.d. samples any output of a differentially-private algorithm can be used in subsequent analysis while controlling the risk of overfitting, even beyond the setting of statistical queries studied in [7]. A key property of differential privacy in this context is that it composes adaptively: namely if each of the algorithms used by the analyst is differentially private, then the whole procedure will be differentially private (albeit with worse privacy parameters). Therefore, one way to avoid overfitting in the adaptive setting is to use algorithms that satisfy (sufficiently strong) guarantees of differential-privacy.

*Description length:* We then show how description length bounds can be applied in the context of guaranteeing generalization in the presence of adaptivity. If the total length of the outputs of algorithms $\mathcal{A}_1, \ldots, \mathcal{A}_{i-1}$ can be described with $k$ bits then there are at most $2^k$ possible values of the input $y_1, \ldots, y_{i-1}$ to $\mathcal{A}_i$. For each of these individual inputs $\mathcal{A}_i$ generalizes with probability $1 - \beta_i$. Taking a union bound over failure probabilities implies generalization with probability at least $1 - 2^k \beta_i$. Occam's Razor famously implies that shorter hypotheses have lower generalization error. Our observation is that shorter hypotheses (and the results of analysis more generally) are also better in the adaptive setting since they reveal less about the dataset and lead to better generalization of *subsequent* analyses. Note that this result makes no assumptions about the data distribution $\mathcal{D}$. In the full versionwe also show that description length-based analysis suffices for obtaining an algorithm (albeit not an efficient one) that can answer an exponentially large number of adaptively chosen statistical queries. This provides an alternative proof for one of the results in [7].

*Approximate max-information:* Our main technical contribution is the introduction and analysis of a new information-theoretic measure, which unifies the generalization arguments that come from both differential privacy and description length, and that quantifies how much information has been learned about the data by the analyst. Formally, for jointly distributed random variables $(\boldsymbol{S}, \boldsymbol{Y})$, the max-information is the maximum of the logarithm of the factor by which uncertainty about $\boldsymbol{S}$ is reduced given the value of $\boldsymbol{Y}$, namely $I_\infty(\boldsymbol{S}, \boldsymbol{Y}) \doteq \log \max \frac{\mathbb{P}[\boldsymbol{S}=S \mid \boldsymbol{Y}=y]}{\mathbb{P}[\boldsymbol{S}=S]}$, where the maximum is taken over all $S$ in the support of $\boldsymbol{S}$ and $y$ in the support $\boldsymbol{Y}$. Approximate max-information is a relaxation of max-information. In our use, $\boldsymbol{S}$ denotes a dataset drawn randomly from the distribution $\mathcal{D}$ and $\boldsymbol{Y}$ denotes the output of a (possibly randomized) algorithm on $\boldsymbol{S}$. We prove that approximate max-information has the following properties

- An upper bound on (approximate) max-information gives generalization guarantees.
- Differentially private algorithms have low max-information for any distribution $\mathcal{D}$ over datasets. A stronger bound holds for approximate max-information on i.i.d. datasets. These bounds apply only to so-called pure differential privacy (the $\delta = 0$ case).
- Bounds on the description length of the output of an algorithm give bounds on the approximate max-information of the algorithm for any $\mathcal{D}$.
- Approximate max-information composes adaptively.

Composition properties of approximate max-information imply that one can easily obtain generalization guarantees for adaptive sequences of algorithms, some of which are differentially private, and others of which have outputs with short description length. These properties also imply that differential privacy can be used to control generalization for any distribution $\mathcal{D}$ over datasets, which extends its generalization guarantees beyond the restriction to datasets drawn i.i.d. from a fixed distribution, as in [7].

We remark that (pure) differential privacy and description length are otherwise incomparable. Bounds on max-information or differential privacy of an algorithm can, however, be translated to bounds on *randomized description length* for a different algorithm with statistically indistinguishable output. Here we say that a randomized algorithm has randomized description length of $k$ if for every fixing of the algorithm's random bits, it has description length of $k$. Details of these results and additional discussion appear in Section 2 and the full version.

## 1.2 Related Work

This work complements [7] where we initiated the formal study of adaptivity in data analysis. The primary focus of [7] is the problem of answering adaptively chosen statistical queries. The main technique is a strong connection between differential privacy and generalization: differential privacy

guarantees that the distribution of outputs does not depend too much on any one of the data samples, and thus, differential privacy gives a strong stability guarantee that behaves well under adaptive data analysis. The link between generalization and approximate differential privacy made in [7] has been subsequently strengthened, both qualitatively — by [1], who make the connection for a broader range of queries — and quantitatively, by [14] and [1], who give tighter quantitative bounds. These papers, among other results, give methods for accurately answering exponentially (in the dataset size) many adaptively chosen queries, but the algorithms for this task are not efficient. It turns out this is for fundamental reasons – Hardt and Ullman [11] and Steinke and Ullman [19] prove that, under cryptographic assumptions, no efficient algorithm can answer more than quadratically many statistical queries chosen adaptively by an adversary who knows the true data distribution.

The classical approach in theoretical machine learning to ensure that empirical estimates generalize to the underlying distribution is based on the various notions of complexity of the set of functions output by the algorithm, most notably the VC dimension. If one has a sample of data large enough to guarantee generalization for all functions in some class of bounded complexity, then it does not matter whether the data analyst chooses functions in this class adaptively or non-adaptively. Our goal, in contrast, is to prove generalization bounds *without* making any assumptions about the class from which the analyst can output functions.

An important line of work [3, 15, 18] establishes connections between the *stability* of a learning algorithm and its ability to generalize. Stability is a measure of how much the output of a learning algorithm is perturbed by changes to its input. It is known that certain stability notions are necessary and sufficient for generalization. Unfortunately, the stability notions considered in these prior works do not compose in the sense that running multiple stable algorithms sequentially and adaptively may result in a procedure that is not stable. The measure we introduce in this work (max information), like differential privacy, has the strength that it enjoys adaptive composition guarantees. This makes it amenable to reasoning about the generalization properties of adaptively applied sequences of algorithms, while having to analyze only the individual components of these algorithms. Connections between stability, empirical risk minimization and differential privacy in the context of learnability have been recently explored in [21].

Numerous techniques have been developed by statisticians to address common special cases of adaptive data analysis. Most of them address a single round of adaptivity such as variable selection followed by regression on selected variables or model selection followed by testing and are optimized for specific inference procedures (the literature is too vast to adequately cover here, see Ch. 7 in [12] for a textbook introduction and [20] for a survey of some recent work). In contrast, our framework addresses multiple stages of adaptive decisions, possible lack of a predetermined analysis protocol and is not restricted to any specific procedures.

Finally, inspired by our work, Blum and Hardt [2] showed how to reuse the holdout set to maintain an accurate leaderboard in a machine learning competition that allows the participants to submit adaptively chosen models in the process of the competition (such as those organized by Kaggle Inc.). Their analysis also relies on the description length-based technique we used to analyze SparseValidate.

## 2   Max-Information

**Preliminaries:** In the discussion below $\log$ refers to binary logarithm and $\ln$ refers to the natural logarithm. For two random variables $\boldsymbol{X}$ and $\boldsymbol{Y}$ over the same domain $\mathcal{X}$ the max-divergence of $\boldsymbol{X}$ from $\boldsymbol{Y}$ is defined as $D_\infty(\boldsymbol{X}\|\boldsymbol{Y}) = \log \max_{x \in \mathcal{X}} \frac{\mathbb{P}[\boldsymbol{X}=x]}{\mathbb{P}[\boldsymbol{Y}=x]}$. $\delta$-approximate max-divergence is defined as

$$D_\infty^\delta(\boldsymbol{X}\|\boldsymbol{Y}) = \log \max_{\mathcal{O} \subseteq \mathcal{X},\, \mathbb{P}[\boldsymbol{X} \in \mathcal{O}] > \delta} \frac{\mathbb{P}[\boldsymbol{X} \in \mathcal{O}] - \delta}{\mathbb{P}[\boldsymbol{Y} \in \mathcal{O}]}.$$

**Definition 1.** *[9, 8] A randomized algorithm $\mathcal{A}$ with domain $\mathcal{X}^n$ for $n > 0$ is $(\varepsilon, \delta)$-differentially private if for all pairs of datasets that differ in a single element $S, S' \in \mathcal{X}^n$: $D_\infty^\delta(\mathcal{A}(S)\|\mathcal{A}(S')) \leq \log(e^\varepsilon)$. The case when $\delta = 0$ is sometimes referred to as* pure *differential privacy, and in this case we may say simply that $\mathcal{A}$ is $\varepsilon$-differentially private.*

Consider two algorithms $\mathcal{A} : \mathcal{X}^n \to \mathcal{Y}$ and $\mathcal{B} : \mathcal{X}^n \times \mathcal{Y} \to \mathcal{Y}'$ that are composed adaptively and assume that for every fixed input $y \in \mathcal{Y}$, $\mathcal{B}$ generalizes for all but fraction $\beta$ of datasets. Here we are speaking of generalization informally: our definitions will support any property of input $y \in \mathcal{Y}$

and dataset $S$. Intuitively, to preserve generalization of $\mathcal{B}$ we want to make sure that the output of $\mathcal{A}$ does not reveal too much information about the dataset $S$. We demonstrate that this intuition can be captured via a notion of *max-information* and its relaxation *approximate max-information*.

For two random variables $\boldsymbol{X}$ and $\boldsymbol{Y}$ we use $\boldsymbol{X} \times \boldsymbol{Y}$ to denote the random variable obtained by drawing $\boldsymbol{X}$ and $\boldsymbol{Y}$ independently from their probability distributions.

**Definition 2.** *Let $\boldsymbol{X}$ and $\boldsymbol{Y}$ be jointly distributed random variables. The max-information between $\boldsymbol{X}$ and $\boldsymbol{Y}$ is defined as $I_\infty(\boldsymbol{X}; \boldsymbol{Y}) = D_\infty((\boldsymbol{X}, \boldsymbol{Y}) \| \boldsymbol{X} \times \boldsymbol{Y})$. The $\beta$-approximate max-information is defined as $I_\infty^\beta(\boldsymbol{X}; \boldsymbol{Y}) = D_\infty^\beta((\boldsymbol{X}, \boldsymbol{Y}) \| \boldsymbol{X} \times \boldsymbol{Y})$.*

In our use $(\boldsymbol{X}, \boldsymbol{Y})$ is going to be a joint distribution $(\boldsymbol{S}, \mathcal{A}(\boldsymbol{S}))$, where $\boldsymbol{S}$ is a random $n$-element dataset and $\mathcal{A}$ is a (possibly randomized) algorithm taking a dataset as an input.

**Definition 3.** *We say that an algorithm $\mathcal{A}$ has $\beta$-approximate max-information of $k$ if for every distribution $\mathcal{S}$ over $n$-element datasets, $I_\infty^\beta(\boldsymbol{S}; \mathcal{A}(\boldsymbol{S})) \leq k$, where $\boldsymbol{S}$ is a dataset chosen randomly according to $\mathcal{S}$. We denote this by $I_\infty^\beta(\mathcal{A}, n) \leq k$.*

An immediate corollary of our definition of approximate max-information is that it controls the probability of "bad events" that can happen as a result of the dependence of $\mathcal{A}(S)$ on $S$.

**Theorem 4.** *Let $\boldsymbol{S}$ be a random dataset in $\mathcal{X}^n$ and $\mathcal{A}$ be an algorithm with range $\mathcal{Y}$ such that for some $\beta \geq 0$, $I_\infty^\beta(\boldsymbol{S}; \mathcal{A}(\boldsymbol{S})) = k$. Then for any event $\mathcal{O} \subseteq \mathcal{X}^n \times \mathcal{Y}$,*

$$\mathbb{P}[(\boldsymbol{S}, \mathcal{A}(\boldsymbol{S})) \in \mathcal{O}] \leq 2^k \cdot \mathbb{P}[\boldsymbol{S} \times \mathcal{A}(\boldsymbol{S}) \in \mathcal{O}] + \beta.$$

*In particular, $\mathbb{P}[(\boldsymbol{S}, \mathcal{A}(\boldsymbol{S})) \in \mathcal{O}] \leq 2^k \cdot \max_{y \in \mathcal{Y}} \mathbb{P}[(\boldsymbol{S}, y) \in \mathcal{O}] + \beta$.*

We remark that mutual information between $\boldsymbol{S}$ and $\mathcal{A}(\boldsymbol{S})$ would not suffice for ensuring that bad events happen with tiny probability. For example mutual information of $k$ allows $\mathbb{P}[(\boldsymbol{S}, \mathcal{A}(\boldsymbol{S})) \in \mathcal{O}]$ to be as high as $k/(2\log(1/\delta))$, where $\delta = \mathbb{P}[\boldsymbol{S} \times \mathcal{A}(\boldsymbol{S}) \in \mathcal{O}]$.

Approximate max-information satisfies the following adaptive composition property:

**Lemma 5.** *Let $\mathcal{A} : \mathcal{X}^n \to \mathcal{Y}$ be an algorithm such that $I_\infty^{\beta_1}(\mathcal{A}, n) \leq k_1$, and let $\mathcal{B} : \mathcal{X}^n \times \mathcal{Y} \to \mathcal{Z}$ be an algorithm such that for every $y \in \mathcal{Y}$, $\mathcal{B}(\cdot, y)$ has $\beta_2$-approximate max-information $k_2$. Let $\mathcal{C} : \mathcal{X}^n \to \mathcal{Z}$ be defined such that $\mathcal{C}(S) = \mathcal{B}(\boldsymbol{S}, \mathcal{A}(\boldsymbol{S}))$. Then $I_\infty^{\beta_1 + \beta_2}(\mathcal{C}, n) \leq k_1 + k_2$.*

**Bounds on Max-information:** Description length $k$ gives the following bound on max-information.

**Theorem 6.** *Let $\mathcal{A}$ be a randomized algorithm taking as an input an $n$-element dataset and outputting a value in a finite set $\mathcal{Y}$. Then for every $\beta > 0$, $I_\infty^\beta(\mathcal{A}, n) \leq \log(|\mathcal{Y}|/\beta)$.*

Next we prove a simple bound on max-information of differentially private algorithms that applies to all distributions over datasets.

**Theorem 7.** *Let $\mathcal{A}$ be an $\epsilon$-differentially private algorithm. Then $I_\infty(\mathcal{A}, n) \leq \log e \cdot \epsilon n$.*

Finally, we prove a stronger bound on approximate max-information for datasets consisting of i.i.d. samples using the technique from [7].

**Theorem 8.** *Let $\mathcal{A}$ be an $\varepsilon$-differentially private algorithm with range $\mathcal{Y}$. For a distribution $\mathcal{P}$ over $\mathcal{X}$, let $\boldsymbol{S}$ be a random variable drawn from $\mathcal{P}^n$. Let $\boldsymbol{Y} = \mathcal{A}(\boldsymbol{S})$ denote the random variable output by $\mathcal{A}$ on input $\boldsymbol{S}$. Then for any $\beta > 0$, $I_\infty^\beta(\boldsymbol{S}; \mathcal{A}(\boldsymbol{S})) \leq \log e(\varepsilon^2 n/2 + \varepsilon\sqrt{n \ln(2/\beta)/2})$.*

One way to apply a bound on max-information is to start with a concentration of measure result which ensures that the estimate of predictor's accuracy is correct with high probability when the predictor is chosen independently of the samples. For example for a loss function with range $[0, 1]$, Hoeffding's bound implies that for a dataset consisting of i.i.d. samples the empirical estimate is not within $\tau$ of the true accuracy with probability $\leq 2e^{-2\tau^2 n}$. Now, given a bound of $\log e \cdot \tau^2 n$ on $\beta$-approximate information of the algorithm that produces the estimator, Thm. 4 implies that the produced estimate is not within $\tau$ of the true accuracy with probability $\leq 2^{\log e \cdot \tau^2 n} \cdot 2e^{-2\tau^2 n} + \beta \leq 2e^{-\tau^2 n} + \beta$. Thm. 7 implies that any $\tau^2$-differentially private algorithm has max-information of at most $\log e \cdot \tau^2 n$. For a dataset consisting of i.i.d. samples Thm. 8 implies that a $\tau$-differentially private algorithm has $\beta$-approximate max-information of $1.25 \log e \cdot \tau^2 n$ for $\beta = 2e^{-\tau^2 n}$.

## 3 Reusable Holdout

We describe two simple algorithms that enable validation of analyst's queries in the adaptive setting.
**Thresholdout:** Our first algorithm Thresholdout follows the approach in [7] where differentially private algorithms are used to answer adaptively chosen statistical queries. This approach can also be applied to any low-sensitivity functions of the dataset but for simplicity we present the results for statistical queries. Here we address an easier problem in which the analyst's queries only need to be answered when they overfit. Also, unlike in [7], the analyst has full access to the training set and the holdout algorithm only prevents overfitting to holdout dataset. As a result, unlike in the general query answering setting, our algorithm can efficiently validate an exponential in $n$ number of queries as long as a relatively small number of them overfit.

For a function $\phi : \mathcal{X} \to \mathbb{R}$ and a dataset $S = (x_1, \ldots, x_n)$, let $\mathcal{E}_S[\phi] \doteq \frac{1}{n} \sum_{i=1}^n \phi(x_i)$. Thresholdout is given access to the training dataset $S_t$ and holdout dataset $S_h$ and a budget limit $B$. It allows any query of the form $\phi : \mathcal{X} \to [0,1]$ and its goal is to provide an estimate of $\mathcal{P}[\phi]$. To achieve this the algorithm gives an estimate of $\mathcal{E}_{S_h}[\phi]$ in a way that prevents overfitting of functions generated by the analyst to the holdout set. In other words, responses of Thresholdout are designed to ensure that, with high probability, $\mathcal{E}_{S_h}[\phi]$ is close to $\mathcal{P}[\phi]$ and hence an estimate of $\mathcal{E}_{S_h}[\phi]$ gives an estimate of the true expectation $\mathcal{P}[\phi]$.

Given a function $\phi$, Thresholdout first checks if the difference between the average value of $\phi$ on the training set $S_t$ (or $\mathcal{E}_{S_t}[\phi]$) and the average value of $\phi$ on the holdout set $S_h$ (or $\mathcal{E}_{S_h}[\phi]$) is below a certain threshold $T + \eta$. Here, $T$ is a fixed number such as $0.01$ and $\eta$ is a Laplace noise variable whose standard deviation needs to be chosen depending on the desired guarantees (The Laplace distribution is a symmetric exponential distribution.) If the difference is below the threshold, then the algorithm returns $\mathcal{E}_{S_t}[\phi]$. If the difference is above the threshold, then the algorithm returns $\mathcal{E}_{S_h}[\phi] + \xi$ for another Laplacian noise variable $\xi$. Each time the difference is above threshold the "overfitting" budget $B$ is reduced by one. Once it is exhausted, Thresholdout stops answering queries. We provide the pseudocode of Thresholdout below.

---

**Input:** Training set $S_t$, holdout set $S_h$, threshold $T$, noise rate $\sigma$, budget $B$

1. sample $\gamma \sim \mathrm{Lap}(2 \cdot \sigma)$; $\hat{T} \leftarrow T + \gamma$
2. **For** each query $\phi$ **do**
   (a) **if** $B < 1$ output "$\perp$"
   (b) **else**
       i. sample $\eta \sim \mathrm{Lap}(4 \cdot \sigma)$
       ii. **if** $|\mathcal{E}_{S_h}[\phi] - \mathcal{E}_{S_t}[\phi]| > \hat{T} + \eta$
           A. sample $\xi \sim \mathrm{Lap}(\sigma)$, $\gamma \sim \mathrm{Lap}(2 \cdot \sigma)$
           B. $B \leftarrow B - 1$ and $\hat{T} \leftarrow T + \gamma$
           C. output $\mathcal{E}_{S_h}[\phi] + \xi$
       iii. **else** output $\mathcal{E}_{S_t}[\phi]$.

---

We now establish the formal generalization guarantees that Thresholdout enjoys.

**Theorem 9.** *Let $\beta, \tau > 0$ and $m \geq B > 0$. We set $T = 3\tau/4$ and $\sigma = \tau/(96 \ln(4m/\beta))$. Let $\boldsymbol{S}$ denote a holdout dataset of size $n$ drawn i.i.d. from a distribution $\mathcal{P}$ and $S_t$ be any additional dataset over $\mathcal{X}$. Consider an algorithm that is given access to $S_t$ and adaptively chooses functions $\phi_1, \ldots, \phi_m$ while interacting with Thresholdout which is given datasets $\boldsymbol{S}, S_t$ and values $\sigma, B, T$. For every $i \in [m]$, let $\boldsymbol{a}_i$ denote the answer of Thresholdout on function $\phi_i : \mathcal{X} \to [0,1]$. Further, for every $i \in [m]$, we define the counter of overfitting $\boldsymbol{Z}_i \doteq |\{j \leq i : |\mathcal{P}[\phi_j] - \mathcal{E}_{S_t}[\phi_j]| > \tau/2\}|$. Then*

$$\mathbb{P}\left[\exists i \in [m], \boldsymbol{Z}_i < B \ \& \ |\boldsymbol{a}_i - \mathcal{P}[\phi_i]| \geq \tau\right] \leq \beta$$

*whenever $n \geq n_0 = O\left(\frac{\ln(m/\beta)}{\tau^2}\right) \cdot \min\{B, \sqrt{B \ln(\ln(m/\beta)/\tau)}\}$.*

**SparseValidate:** We now present a general algorithm for validation on the holdout set that can validate many arbitrary queries as long as few of them fail the validation. More formally, our

algorithm allows the analyst to pick any Boolean function of a dataset $\psi$ (or even any algorithm that outputs a single bit) and provides back the value of $\psi$ on the holdout set $\psi(S_h)$. SparseValidate has a budget $m$ for the total number of queries that can be asked and budget $B$ for the number of queries that returned $1$. Once either of the budgets is exhausted, no additional answers are given. We now give a general description of the guarantees of SparseValidate.

**Theorem 10.** *Let $\boldsymbol{S}$ denote a randomly chosen holdout set of size $n$. Let $\mathcal{A}$ be an algorithm that is given access to SparseValidate$(m, B)$ and outputs queries $\psi_1, \ldots, \psi_m$ such that each $\psi_i$ is in some set $\Psi_i$ of functions from $\mathcal{X}^n$ to $\{0, 1\}$. Assume that for every $i \in [m]$ and $\psi_i \in \Psi_i$, $\mathbb{P}[\psi_i(\boldsymbol{S}) = 1] \leq \beta_i$. Let $\boldsymbol{\psi}_i$ be the random variable equal to the $i$'th query of $\mathcal{A}$ on $\boldsymbol{S}$. Then $\mathbb{P}[\boldsymbol{\psi}_i(\boldsymbol{S}) = 1] \leq \ell_i \cdot \beta_i$, where $\ell_i = \sum_{j=0}^{\min\{i-1, B\}} \binom{i}{j} \leq m^B$.*

In this general formulation it is the analyst's responsibility to use the budgets economically and pick query functions that do not fail validation often. At the same time, SparseValidate ensures that (for the appropriate values of the parameters) the analyst can think of the holdout set as a fresh sample for the purposes of validation. Hence the analyst can pick queries in such a way that failing the validation reliably indicates overfitting. An example of the application of SparseValidate for answering statistical and low-sensitivity queries that is based on our analysis can be found in [1]. The analysis of generalization on the holdout set in [2] and the analysis of the Median Mechanism we give in the full version also rely on this sparsity-based technique.

**Experiments:** In our experiment the analyst is given a $d$-dimensional labeled data set $S$ of size $2n$ and splits it randomly into a training set $S_t$ and a holdout set $S_h$ of equal size. We denote an element of $S$ by a tuple $(x, y)$ where $x$ is a $d$-dimensional vector and $y \in \{-1, 1\}$ is the corresponding class label. The analyst wishes to select variables to be included in her classifier. For various values of the number of variables to select $k$, she picks $k$ variables with the largest absolute correlations with the label. However, she verifies the correlations (with the label) on the holdout set and uses only those variables whose correlation agrees in sign with the correlation on the training set and both correlations are larger than some threshold in absolute value. She then creates a simple linear threshold classifier on the selected variables using only the signs of the correlations of the selected variables. A final test evaluates the classification accuracy of the classifier on both the training set and the holdout set.

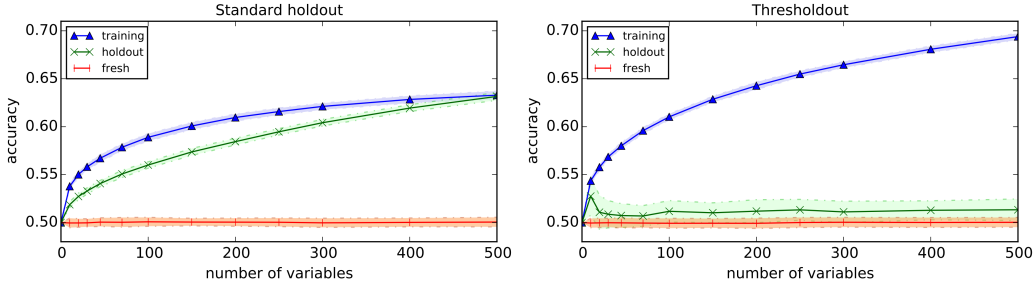

In our first experiment, each attribute of $x$ is drawn independently from the normal distribution $N(0, 1)$ and we choose the class label $y \in \{-1, 1\}$ uniformly at random so that there is no correlation between the data point and its label. We chose $n = 10,000$, $d = 10,000$ and varied the number of selected variables $k$. In this scenario no classifier can achieve true accuracy better than $50\%$. Nevertheless, reusing a standard holdout results in reported accuracy of over $63\%$ for $k = 500$ on both the training set and the holdout set (the standard deviation of the error is less than $0.5\%$). The average and standard deviation of results obtained from $100$ independent executions of the experiment are plotted above. For comparison, the plot also includes the accuracy of the classifier on another fresh data set of size $n$ drawn from the same distribution. We then executed the same algorithm with our reusable holdout. Thresholdout was invoked with $T = 0.04$ and $\tau = 0.01$ explaining why the accuracy of the classifier reported by Thresholdout is off by up to $0.04$ whenever the accuracy on the holdout set is within $0.04$ of the accuracy on the training set. We also used Gaussian noise instead of Laplacian noise as it has stronger concentration properties. Thresholdout prevents the algorithm from overfitting to the holdout set and gives a valid estimate of classifier accuracy. Additional experiments and discussion are presented in the full version.

## Footnotes

*See [6] for the full version of this work.

†Part of this work done while visiting the Simons Institute, UC Berkeley.

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
