[Reviews · NeurIPS 2015]

Submitted by Assigned_Reviewer_1

This paper advances understanding of generalization error in adaptive data analysis by introduction the notion of max-information, establishing connections to differential privacy- and description length-based adaptive data analysis, and developing new algorithms. Max-information is a natural quantity for this problem and I suspect that this development will enable new algorithms and new approaches for correct adaptive data analysis, such as the description length-based approach in the paper.

Quality: The paper studies an important problem and significantly advances our understanding. The paper has a good blend of theory -- with the analysis of the max-information notion -- and practice -- with two new algorithms and empirical results.

Clarity: I think one minor shortcoming of the paper is that the main text packs in too much content into too little space. The full version of the paper is much more readable but maybe there are some results that can be condensed or omitted in the main text? I found the introduction verbose and believe it can be significantly condensed.

Another thing that I think would be helpful is a comparison of the main Theorems to those in [6]. I think the discussion following Theorem 8 is useful but I would like to have a more formal comparison. Looking at [6], I think the right results to compare are Theorem 9 in [6] with Corollary 20 here. Qualitatively the bounds seem to be similar -- for sensitive 1/n functions, setting the differential privacy parameter to be tau gives deviation bounds analogous to Hoeffding's bound in both cases. If I understand that correctly, I think it is important to point out.

More generally, being very explicit about the key contributions of the paper would aid readability.

Originality: Original.

Significance: Significant.

Some minor comments: 1. In Definition 3 I_\infty^\beta((S, \Acal(S)) should be I_\infty^\beta(S; \Acal(S)).

2. In the experiment section (in the main text) you mention that you also use Gaussian noise instead of Laplace noise. Is that shown in the figure? Or is Laplace noise shown there?
Summary: This paper studies the important problem of generalization in adaptive data analysis and gives new theoretical insights and algorithms. I think the results in this paper are important, and apart from minor issues mostly having to do with many results packed into few pages, I have very few complaints.

Submitted by Assigned_Reviewer_2

# Summary

The paper considers the setting where an analyst has a training set and a hold-out set from the same distribution. The analyst constructs her methods based on the training set, but also repeatedly performs checks on the hold-out set to make important decisions. In general this repeated use of the hold-out set may lead to overfitting to the hold-out set, but if the analyst only performs her checks using the methods introduced in the paper, overfitting is avoided, as shown by the main results: Thms 9 and 10. This extends earlier results based on differentially private methods.

# Positive Points

What is exciting about this line of results, is that they do not a priori restrict the class of functions (say, to a class of limited VC dimension) from which the analyst can select her checks. Otherwise, standard uniform concentration inequalities for VC classes would suffice. See lines 226-232 in the paper.

An interesting point of the paper is that it goes beyond differentially private methods to also include algorithms whose output can be described by a small number of bits. This generalization is based on a unifying quantity called (approximate) max-information. Unlike notions of algorithmic stability, which are known to be closely related to generalization, this max-information allows composing multiple algorithms in a sequence. (See lines 234-243.)

The high-level discussion in the paper is generally well-written.

# Negative Points

In spite of all the positive points mentioned above, I have some significant concerns about Thm 9 (one of the main results), which currently makes me believe that the paper would need a serious revision before it could be accepted.

# Main concerns

Regarding Thm 9: 1. The theorem makes a deterministic statement "for all i such that

a_i \neq \perp", but whether "a_i \neq \perp" is random, because it

depends on the adaptively chosen functions phi_i, on the randomness

in the data, and on the randomness introduced by the algorithm. 2. The in-probability statement bounds the probability that a_i deviates

significantly from the expectation of phi_i for a given fixed i. But

if the analyst adaptively makes decisions based on the output of the

algorithm, then a_i must be close to the expectation of phi_i for all

i simultaneously. Otherwise the analysis might take a wrong turn

somewhere. 3. I have been unable to verify the claim in lines 119,120 of the

introduction, which I paraphrase as: the number of queries m can be

exponential in n as long as the budget B is at most quadratic in n.

In particular, the result does not appear to deteriorate with m,

which I think is because of point 2 above. And in order for the

result to be useful to the analyst, I would expect that tau would go

to 0 with n, for example as tau ~ 1/sqrt(n). But if we let B be

quadratic in n, then having tau go to 0 will make beta go to

infinity, and hence the result becomes vacuous. So I don't know which

parameters to plug into the theorem.

About the experiments: why do you construct a classifier that only uses weights +1 and -1 based on the signs of correlations? Is there a plausible scenario in which someone might do this in practice?

# Minor remarks

As acknowledged by the authors in their proof of Thm 23 in the additional material, the extension to algorithms whose output can be described using a small number of bits can also be obtained from a union bound argument. This is hinted at in lines 171-179 of the introduction, but may not be entirely clear from the discussion there.

# A potential strengthening

Finally, I would like to point out a possible strengthening of the results, which may or may not be useful.

Technically, all results are based on a change of measure from the joint distribution P of a sample and an algorithm's output on that sample to the independent product distribution Q of the two (Thm 4). If P and Q are sufficiently close in terms of "max-information", then we may essentially still treat the sample as fresh even if we have already looked at the output of the algorithm.

It might be of interest to observe that, at least for beta = 0, the bound on max-information can be relaxed. To do this, the max-information may be interpreted as the Renyi divergence D_alpha(P||Q) of order alpha = infinity. Renyi divergence is increasing in alpha (see e.g. [1]), so requiring a bound on alpha=infty is the strongest requirement one can impose. Thm 4 then states that we can change measures from P to Q if D_infty(P||Q) is sufficiently small. It is actually possible to change measures under the weaker requirement that D_alpha(P||Q) is small for any alpha > 1. For alpha=1 we recover regular mutual information, which, as the authors point out, is not strong enough to allow a change of measure.

The proof is essentially a special case of Lemma 1 of [2], but the relation may be hard to see, so allow me to translate. Let X be the indicator random variable, which is 1 if (S,A(S)) in cal{O} and 0 otherwise. Then read Lemma 1 of [2] with E_P[X] instead of L_P(h,f), E_Q[X] instead of L_Q(h,f), and M = max X = 1 to get:

E_P[X] <= ( 2^{D_alpha(P||Q)} * E_Q[X] )^{(alpha-1)/alpha} * M^(1/alpha)

which is equivalent to

P[(S,A(S)) in cal{O}] <= ( 2^{D_alpha(P||Q)} * Q[(S,A(S)) in cal{O}] )^{(alpha-1)/alpha}

As alpha -> infinity, we recover Thm 4, but the result also holds for smaller alpha.

1. Van Erven, Harremoes, "Renyi Divergence and Kullback-Leibler Divergence, IEEE Transactions on Information Theory, 2014. 2. Mansour, Mohri, Rostamizadeh, "Multiple Source Adaptation and the Renyi Divergence", UAI 2009.

# Minor issues (typo's etc)

Line 114: training set -> hold-out set (I believe)

Thm 4: two issues: "k = I_infty^beta(S;A(S)) = k" -> I_infty^beta(S;A(S)) <= k?

Thm 8: The sentence "Let Y = ... on input S." can be removed.

Lines 314-323: the bounds you mention bound the probability that the empirical estimate is more than tau *away* from the true accuracy.

Summary: In spite of many positive points, I have some significant concerns about Thm 9 (one of the main results), which currently makes me believe that the paper would need a serious revision before it could be accepted.

Submitted by Assigned_Reviewer_3

The paper introduces and studies an information-theoretic quantity, approximate max-information, which is used to quantify generalization properties of adaptive data analyses. The output of a data analysis algorithm with low approximate max-information can be regarded as "almost" independent of the data itself, in a particular sense that is useful for reasoning about subsequent analyses of the data that may depend on the algorithm's output. Differentially private algorithms have low approximate max-information, as do algorithms with a small cardinality range, and compositions of algorithms with low approximate max-information also have relatively low approximate max-information. A few applications are given: generalization of differentially private algorithms (Corollaries 19 and 20), a procedure for managing holdout set reuse in machine learning (Thresholdout), and a procedure for multiple hypothesis testing (or activities like that) on the same data (SparseValidate).

I think the paper should be accepted, as the paper gives an interesting perspective on generalization. Although it is similar to algorithmic stability concepts, I think approximate max-information captures a nice property that seems to be more useful and versatile than previous stability concepts (largely because of the composition property). All the applications are very interesting as well.

The paper fails to compare to some earlier work on generalization from learning theory. For instance, both Freund's work on self-bounding learning algorithms and Blum and Langford's work on micro-choice bounds seem particularly relevant. I also had some difficulty seeing the difference between the description length results and previous Occam/MDL-type bounds (again, see work by Blum and Langford).

I think the authors should discuss the interplay between tau and n in Thresholdout. Often, we hope to at least have tau = O(1/\sqrt{n}); this would imply budgets of constant size or smaller.

The results, of course, are still interesting in this case due to the allowance of adaptivity, and the "free" evaluation of "good" functions.

But I think pointing out this quantitative aspect is important for putting the results in perspective.

Some other comments are below: - Page 6, line 289: "k = I_\infty^\beta(S; A(S)) = k" - Page 6, line 318,320: probability statements seem to be inverted. - Page 7, line 338: define $\mathcal E_{S_h}[\phi]$ - Page 7, line 341: do you mean "true expectation $\mathcal P[\phi]$"?

Yet even more comments: - Another reviewer points out some issue with the statement of Theorem 9. Did you mean something like "For all i and t, Pr{ a_i \neq \bot AND |a_i - P[\phi_i]| > T + (t+1)\tau } \leq ..."? This would be good to clear up. - It would also be good to be more explicit about various qualitative claims (e.g., # queries can be exponential in n). The user may not know when the budget will be exhausted, but may want simultaneous validity of all the non-\bot queries. So it seems that you will want to apply Theorem 9 with a union bound over the m queries. (Perhaps something more clever than a straight-up union bound could be done.) - Also, why does it matter that S_t (training set) be an iid sample?

---

Post-rebuttal remarks:

One major point of confusion in the text is Line 119-120. It sounds like we can have m = 2^{cn} and B = cn^2 _simultaneously_ for some positive, but this clearly leads to trivial results in Theorem 9. Perhaps it's fine to use the terms "quadratic" and "exponential" a bit loosely, but I think it should be stated precisely and explicitly at least somewhere in the paper.
Summary: I think the paper should be accepted, as the paper gives an interesting perspective on generalization. Although it is similar to algorithmic stability concepts, I think approximate max-information captures a nice property that seems to be more useful and versatile than previous stability concepts (largely because of the composition property), and all the applications are very interesting.

Author Feedback
Author rebuttal: We thank all reviewers for their valuable comments and careful reading of our paper.

Reviewer 4 regarding Theorem 9:

We apologize for not making the step from the claim in lines 119-120 to our Thm.9 explicit. At the same time both the claims in the introduction and in Thm.9 are correct and we will be sure to clarify the points that confused the reviewer in the revision. Let us now address the specific points:
1. Indeed, the tail bound stated in Theorem 9 (line 365) is for every fixed query i and not for the whole execution. The probability statement is over the randomness in the data, the queries and our algorithm (so both \phi_i and a_i are random variables). To obtain a tail bound for the maximum error over all queries (which ensures that no "wrong turns" are taken in the whole execution), we may directly apply the union bound over all m queries. The union bound introduces a logarithmic dependence on m (the total number of queries) in the error bound. Hence, without the union bound, the theorem does not have any dependence on m. We think that this is an interesting property that required a more careful analysis. Lines 119/120 in the introduction, however, refer to the statement obtained after taking the union bound. Also in the introduction the statement is in terms of the number of times overfitting occurs whereas B measures the number of "\perp" answers. To make those the same (with high probability) the parameter T needs to be large enough introducing another logarithmic factor in m. Since the dependence is logarithmic in m, an exponential number of queries (in the size of the holdout) can be answered before the bound becomes vacuous.
Formally, to apply the union bound, choose \beta = \beta'/2m and t = 8\log(2m/\beta') for a desired error probability \beta'>0, this gives that \Pr\{\forall i: |a_i - {\cal P}[\phi_i]| \le T + (t+1)\tau\} \le \beta'. Note that T needs to be at least \Omega(\tau (\log m/\beta')) to avoid getting "\bot" responses for queries that do not overfit. Beyond that, the larger we choose T, the fewer queries that are above the threshold but also the higher the error is. The log(m) factor enters the error bound through the choice of t and T since the accuracy of the responses equals tau' = T+ (t+1)\tau = O(\tau \log (m/\beta')). Also B will be smaller by a log(m) factor now. To understand how the response accuracy \tau' tends to 0 with n = |S_h|, simply solve for \tau in the definition of B. \tau = O(B^{1/4}\log(m/(\tau \beta'))^{1/4} / \sqrt{n}).
Using tau' = O(\tau \log(m/\beta')) we get that
\tau' = O(B^{1/4}\log(mn/(\tau\beta'))^{1/4} \log(m/\beta') / \sqrt{n}).

So, we see that, up to a logarithmic factor, \tau' indeed tends to zero as 1/\sqrt{n}. We can also make B nearly quadratic in n and \tau' will still tend to zero, albeit at a slower rate. It is true that for B = n^2, \tau no longer tend to zero. When we said "quadratic" in the prose, we generally omitted "up to logarithmic factors".

2. Regarding the suggestion: Thank you. This is an interesting generalization and we will look into its possible applications in this context.

3. The learning algorithm is chosen to be very simple and solely for the illustration. At the same time, each variable or its negation can be thought of as a "simple" predictor. Unweighted voting of simple predictors is used in such popular methods as bagging.

Reviewer 2:

1. Regarding the interplay of \tau and n, please see the above paragraph. You are absolutely right that for \tau ~ 1/\sqrt{n} (up to a logarithmic factor), we only get a constant budget B. Determining the optimal quantitative bounds is a wonderful open question! We will add an extended discussion of the parameter interplay to the main body of the paper.
2. We will clarify the relationship to self-bounding and microchoice approaches you mentioned (and of which we are aware). There is some overlap in the basic tools we use: description length and statistical queries. At the same time the works are conceptually very different. The primary goal in those works is to derive data-dependent generalization bounds for a given algorithm. The goal of our work is to give new algorithms which have better generalization properties in the adaptive setting.

Reviewer 1:

Corollary 20 is analogous to McDiarmid's inequality in terms of generality, while Theorem 9 in [6] is analogous to Hoeffding's bound. Corollary 20 also follows easily from results in [6] and we include it primarily as an example application of the new and more general max-information-based technique.